# Cloning of Cold-Adapted Dextranase and Preparation of High Degree Polymerization Isomaltooligosaccharide

**Huanyu Wang** [1,2], **Qianru Lin** [1,2], **Dongxue Dong** [1,2], **Yingying Xu** [1,2], **Mingwang Liu** [1,2], **Jing Lu** [1,2,*], **Mingsheng Lyu** [1,2] **and Shujun Wang** [1,2,*]

1   Jiangsu Key Laboratory of Marine Bioresources and Environment/Jiangsu Key Laboratory of Marine Biotechnology, Jiangsu Ocean University, Lianyungang 222005, China; wanghuanyuhk@163.com (H.W.); lqr.carpedime@outlook.com (Q.L.); dongdongxueddx@163.com (D.D.); ymiabooo@163.com (Y.X.); ironman1307@163.com (M.L.); mslyu@jou.edu.cn (M.L.)
2   Co-Innovation Center of Jiangsu Marine Bio-Industry Technology, Jiangsu Ocean University, Lianyungang 222005, China
*   Correspondence: jinglu@jou.edu.cn (J.L.); sjwang@jou.edu.cn (S.W.)

**Abstract:** Intestinal diseases are mainly caused by a decrease in the relative abundance of probiotics and an increase in the number of pathogenic bacteria due to dysbiosis of the intestinal flora. High degree polymerization isomaltooligosaccharide (IMO) can promote probiotic metabolism and proliferation. In this study, the dextranase (PsDex1711) gene of marine bacterial *Pseudarthrobacter* sp. RN22 was cloned and expressed in *Escherichia coli* BL21 (DE3). The optimal pH and temperature of the dextranase were 6.0 and 30 °C, respectively, showing the highest stability at 20 °C. The dextran T70 could be hydrolyzed to produce IMO3, IMO4, IMO5, and IMO6 with a high degree of polymerization. The hydrolysate of 1 mg/mL could significantly promote the growth of *Lactobacillus* and *Bifidobacterium* after 12 h culture and the formation of biofilms by 58.2%. The hydrolysates could promote the proliferation of probiotics. Furthermore, the $IC_{50}$ of scavenging rate of DPPH, hydroxyl radical, and superoxide anion was less than 20 mg/mL. This study provides a crucial theoretical basis for the application of dextranase such as pharmaceutical and food industries.

**Keywords:** cold-adapted dextranase; cloning expression; isomaltooligosaccharide; prebiotics





## 1. Introduction

Intestinal diseases are frequently occurring diseases, with a total incidence rate is approximately 20% of the total population. Chronic diarrhea, chronic liver disease, and intestinal cancer caused by intestinal flora imbalance are some examples of intestinal diseases [1]. The imbalance of intestinal flora is manifested through a reduction in bacterial diversity, relative abundance of butyrate-producing bacteria, and several other beneficial microorganisms (such as *Lactobacillus* and *Bifidobacterium*), but an increase in the number of some conditional pathogens, including anaerobic bacteria and *Bacteroides* [2]. Prebiotics have attracted widespread attention as being organic substances, they can selectively promote the metabolism and proliferation of probiotics without being digested and absorbed by the host, thereby improving host health [3].

Isomaltooligosaccharides (IMOs) have been widely added to food ingredients as a prebiotics [4]. IMO is an oligosaccharide of isomaltose, panose, isomaltotriose, and above tetrasaccharide, which is composed of a general term for a group of oligosaccharides consisting of 2 to 6 glucose groups bound to each other by alpha-(1,6) glucosidic linkages [5]. IMOs are characterized by low sweetness, crystallization resistance, low viscosity, and prebiotics and are used as a functional ingredient in cosmetics, pharmaceutical, and food industries to improve intestinal health, prevent dental caries, and promote mineral absorption and cholesterol regulation [6].

Traditionally, IMO production from starch involves multiple steps, including liquefaction, two-step saccharification, and purification. At least five enzymes are required to obtain the required IMO structure through this production method [7]. Various enzymes from the hydrolase and transferase families have been used to produce IMOs from disaccharide, oligosaccharide and polysaccharide substrates. These enzymes can produce various IMO structures and have a great potential in reducing production costs [8]. Dextranase (EC 3.2.1.11) can specifically hydrolyze dextran alpha-(1,6) glucosidic linkages and generate D-glucose (D-GL), isomaltose (IMO2), isomaltotriose (IMO3), and various other oligosaccharides [9–11]. Dextranase is mainly produced by bacteria, yeasts, and molds. The properties of dextranase from different sources are obviously different, and this enzyme also has a high substrate specificity [12]. The main dextranase-producing bacterial strains are *Arthrobacter* sp. [13], *Bacillus* sp. [14], and *Pseudoalteromonas* [15]. Dextranase can reduce the viscosity of sugar in the sugar industry, substitute blood plasma in medicine, prevent dental caries and produce prebiotic IMOs [16]. Due to its special living environment, marine bacteria exhibit low temperature resistance, high pressure resistance, and high salt resistance, and are basophilic. which is bound to change their catalytic properties and have more research value in industry and food [17].

However, natural enzymes have the characteristics of low yield, low activity, and poor stability and therefore often cannot meet the requirements of industrial production. Moreover, obtaining high-purity enzymes is difficult [18]. Gene cloning and expression is a common biotechnological method that can be used to solve this problem [19]. Therefore, in this study, the gene coding for dextranase from *Pseudarthrobacter* sp. RN22, a marine bacterium was cloned and expressed, and the pure enzyme was purified using the Ni-NTA nickel column. The properties of the enzyme were studied, and the composition of the hydrolysate—IMO—was analyzed. The antioxidant activity and promoting effect of IMOs on probiotic growth and biofilm formation were also studied. This study provided the research basis for the application of IMOs in medical treatment, and health care, food, and other industries and offered the theoretical basis for the addition of isomaltose to functional foods.

## 2. Results

### 2.1. Sequence Analysis of PsDex1711 Protein

The PsDex1711 gene is 1860-bp long and encodes 620 amino acids. PsDex1711 belongs to the glycoside hydrolase GH49 family. Blast online comparison was performed between PsDex1711 and similar protein sequences in the NCBI database to construct a phylogenetic tree (Figure S1). The similarity between PsDex1711 and Dextranase protein sequences from *Arthrobacter nitrophenolicus* was 93%.

The signal peptide was analyzed using SignalP-5.0 online software. The first 21 amino acids of PsDex1711 are signal peptides. According to the analysis with ExPASy online software, the molecular weight of PsDex1711 is approximately 69,178.79, the molecular formula is $C_{3076}H_{4626}N_{826}O_{958}S_{21}$, the isoelectric point is Pl 5.55, the instability coefficient is 33.48, and hydrophilicity is approximately $-0.466$. TMHMM Server online analysis revealed that PsDex1711 had no transmembrane domain, and 620 amino acids were outside the membrane. DNAMAN V6 software analysis clarified that PsDex1711 is a hydrophilic enzyme, and the first 21 amino acids are signal peptides. After removing the signal peptide, hydrophilicity of PsDex1711 was improved. SOPMA online software, which was used to analyze the secondary structure, revealed that PsDex1711 contains 66 $\alpha$-helices, accounting for 10.65% of the whole structure. Moreover, it has 45 $\beta$-foldings, 310 irregular curls, and 199 extension chains, accounting for 7.26%, 50.00%, and 32.10%, respectively, of the whole structure. The CDD database was used for conservative domain analysis: PsDex1711 has a domain at both the N-terminal and C-terminal. The N-terminal domain contains 192 amino acids (Gly 33-thr 225), and the C-terminal domain contains 125 amino acids (Gly 495-Trp 620).

The sequence of PsDex1711 was analyzed by SWISS-MODEL, and AoDex 6nzs.1.A was used as the template for homology modeling (Figure 1). The results showed that the sequence identity was 69.88%, and the coverage rate reached 94%. The N-terminal

structural domain was a β sandwich structure, and the C-terminal structural domain consisted of a β helix structure. According to the Ramachandran diagram, 85.7% of φ and ψ dihedral angles of the protein were found to be in the optimum region by Procheck, 13.3% in other allowable areas, 0.4% in the maximum allowable area, and only 0.4% in the non allowable area. Errat scored 84.965. Therefore, the distribution ratios and the scores indicated that the model was reliable.

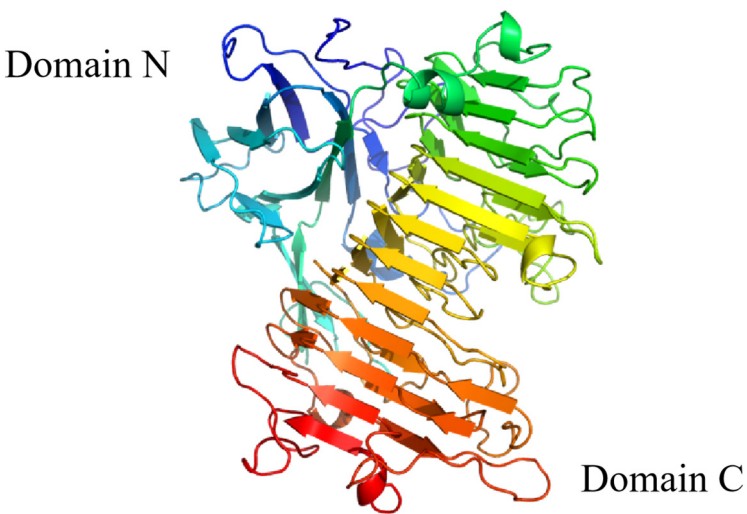

**Figure 1.** The 3D structure model of PsDex1711.

### 2.2. Cloning, Expression, and Purification of the PsDex1711 Gene

The effects of different concentrations of IPTG, induction time, and temperature on PsDex1711 expression are shown in Figure S2. IPTG concentration exhibits enzyme activity at 0.1–1 mM, which is the highest at 0.5 mM. This indicates that too high or too low IPTG concentration is not conducive to PsDex1711 expression. The enzyme activity was the highest at 16 °C, but no activity was observed at 48 °C, indicating that PsDex1711 was only expressed at a low temperature. No significant change in enzyme activity was noted after induction for 24 and 48 h, indicating that the expression did not increase after 24 h, and the optimum fermentation time was 24 h.

After PsDex1711 was successfully expressed under the optimal expression conditions, Ni NTA was used for affinity chromatography purification; the results are presented in Figure 2. After elution with different concentrations of imidazole solution, the miscellaneous protein gradually decreased. It was detected through 10% SDS-PAGE gel electrophoresis. The results showed that 300 mM imidazole had the best elution effect on the miscellaneous protein, and a single band having a molecular weight of approximately 70 kDa was obtained, which was consistent with the results of bioinformatics analysis.

### 2.3. Enzymatic Properties of PsDex1711

The results of PsDex1711 optimal catalytic substrate assay are shown in Table 1, PsDex1711 showed high catalytic activity for dextran of different molecular weights, and the highest enzyme activity was achieved for the substrate dextran T500. At the same time, chitosan with Beta-(1,4) glucosidic linkages, soluble starch and pullulan polysaccharide with alpha-(1,4) and alpha-(1,6) glucosidic linkages displayed no catalytic activity. PsDex1711 specifically hydrolyzes alpha-(1,6) glucosidic linkages. The results of the optimum catalytic substrate assay were the same as those of the wild bacteria.

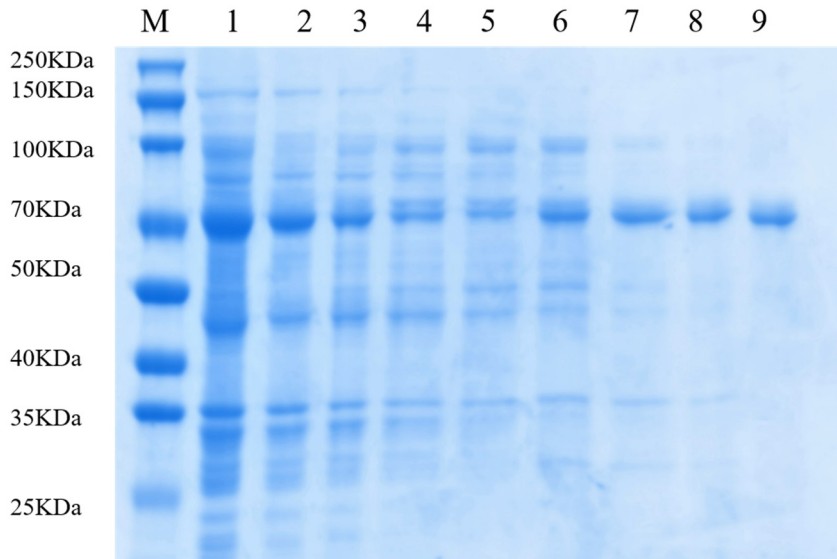

**Figure 2.** 10% SDS-PAGE of PsDex1711 after purification. 1: PsDex1711 crude enzyme solution; 2: Outflow liquid; 3–9: Elute nickel column with 20, 40, 60, 80, 100, 200, and 300 mM imidazole, respectively.

**Table 1.** Optimal catalytic substrate for PsDex1711.

| Substrate | Main Linkages | Relative Activity (%) |
|---|---|---|
| Dextran T20 | $\alpha$-1,6 | 74.60 ± 2.68 |
| Dextran T40 | $\alpha$-1,6 | 79.32 ± 1.33 |
| Dextran T70 | $\alpha$-1,6 | 80.12 ± 3.09 |
| Dextran T500 | $\alpha$-1,6 | 100 ± 2.31 |
| Chitosan | $\beta$-1,4 | 0 |
| Soluble starch | $\alpha$-1,4, $\alpha$-1,6 | 0 |
| Pullulan | $\alpha$-1,4, $\alpha$-1,6 | 0 |

The optimal catalytic temperature and stability determination results of PsDex1711 are shown in Figure 3a,b, the optimum catalytic temperature of PsDex1711 is 30 °C. The enzyme activity decreased rapidly after 40 °C; only 20% enzyme activity was noted at 45 °C, and almost no activity was observed when the temperature continued to rise. The enzyme was observed to have high enzymatic activity at low temperatures: 72.11% and 34.32% were retained at 20 °C and 10 °C, respectively, and nearly 20% were also retained at 0 °C. After holding at 20 °C and 30 °C for 2 h, nearly 95–98% of the enzyme activity was still observed. No enzyme activity was observed after holding at 40 °C and 50 °C for 20 min. Indicating that PsDex1711 has the characteristics of cold adaptability of marine bacteria. Compared to wild bacteria, the optimal catalytic temperature was reduced by 30 °C and higher stability was observed at a low temperature.

The optimal catalytic pH and stability determination results of PsDex1711 are shown in Figure 3c, the optimum catalytic pH of PsDex1711 is 6.0. The enzyme is very sensitive to an acidic environment. Almost no enzyme activity was observed at pH of <5.5, but high enzyme activity was observed in an alkaline environment. When pH was 9.0, 70% enzyme activity was still observed. When pH was 6.0, the stability was the highest, and 40% of the enzyme activity could be maintained when pH was 5.5 or 7.5, indicating that PsDex1711 exhibits alkali resistance and a small amount of weak acid resistance. Compared to wild bacteria, the optimal catalytic pH decreased by 1.5 and the stability also decreased to the acidic range.

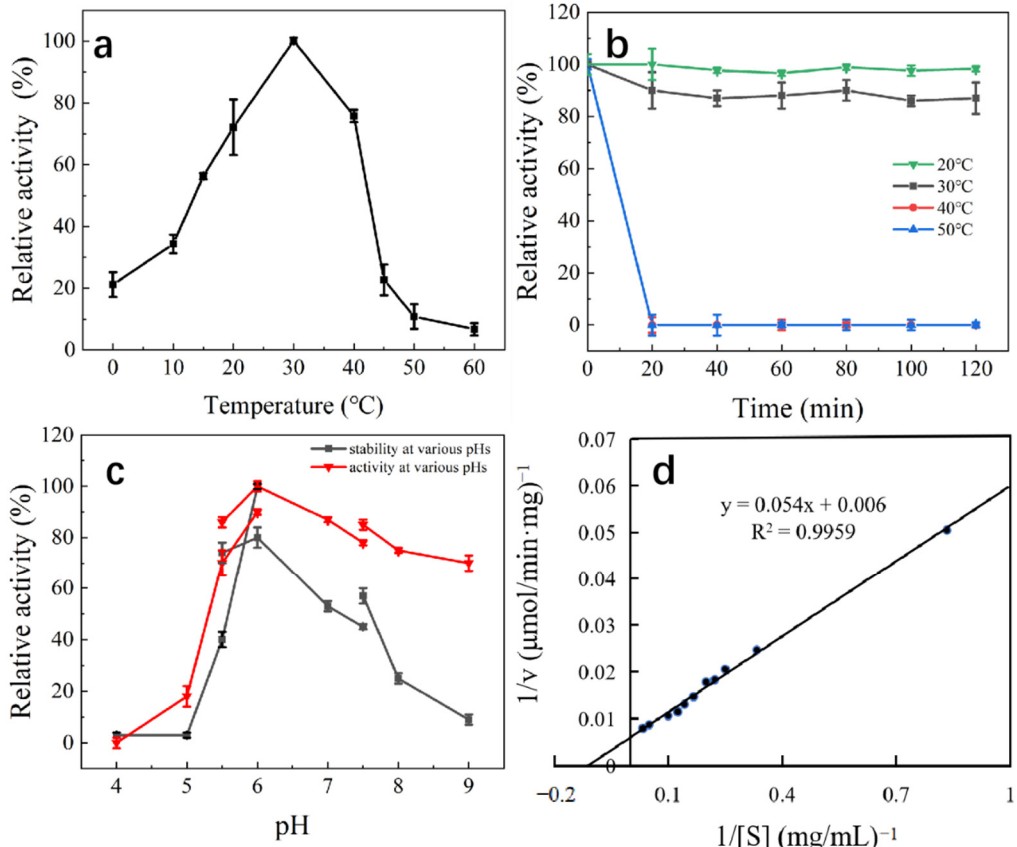

**Figure 3.** Analysis of the enzymatic properties of PsDex1711. (**a**) PsDex1711 optimal catalytic temperature; (**b**) PsDex1711 temperature stability; (**c**) Optimal catalytic pH and stability of PsDex1711; (**d**) Enzyme kinetic constants of PsDex1711.

The enzyme kinetic constants of PsDex1711 were show in Figure 3d. We obtained that the kinetic constant of the enzyme $K_m$ = 8.99 mg/mL and $V_{max}$ = 166.67 μmol/(mg·min). Most dextranases had $K_m$ values between 1–10 mg/mL for the dextran substrate, and the smaller the $K_m$ value, the higher affinity for the substrate [9,20,21]. The results indicated that the enzyme has a fine affinity for the substrate dextran T500.

### 2.4. PsDex1711 Hydrolysate

The TLC results are displayed in Figure 4a. Accordingly, PsDex1711 reacted with dextran T70 for 0.5 h to produce IMO3 and IMO4, 2 h to produce IMO5, and 4 h to produce IMO6. With an increase in the reaction time, the product concentration increased gradually. The HPLC results are displayed in Table 2 and Figure 4b,c shows the standard substances. Within 1 h of reaction, IMO4 accounted for approximately 60% in total, which was twice the content of IMO3. IMO5 was the main component in the reaction for 2–3 h, accounting for 40.50% and 49.18%, respectively. IMO6 was detected after 4 h, and the proportion of IMO3 and IMO4 decreased continuously. The results of analysis of commercial isomaltooligosaccharide is presented in Table S1 and Figure 4d. In CIMO-Zhejiang (CIMO-Z), IMO2 and IMO3 were the main components, accounting for 20.12% and 19.53%, respectively. In CIMO-Shandong (CIMO-S), IMO2 and IMO3 accounted for 25.75% and 25.72%, respectively. Compared with the nonclonal expression, glucose and IMO2 were not found in the product, but a certain amount of isomaltose appeared, and the product changed to a high DP IMO3, which is of great significance for the follow-up study of antioxidant activity of PsDex1711 and its impact on probiotics.

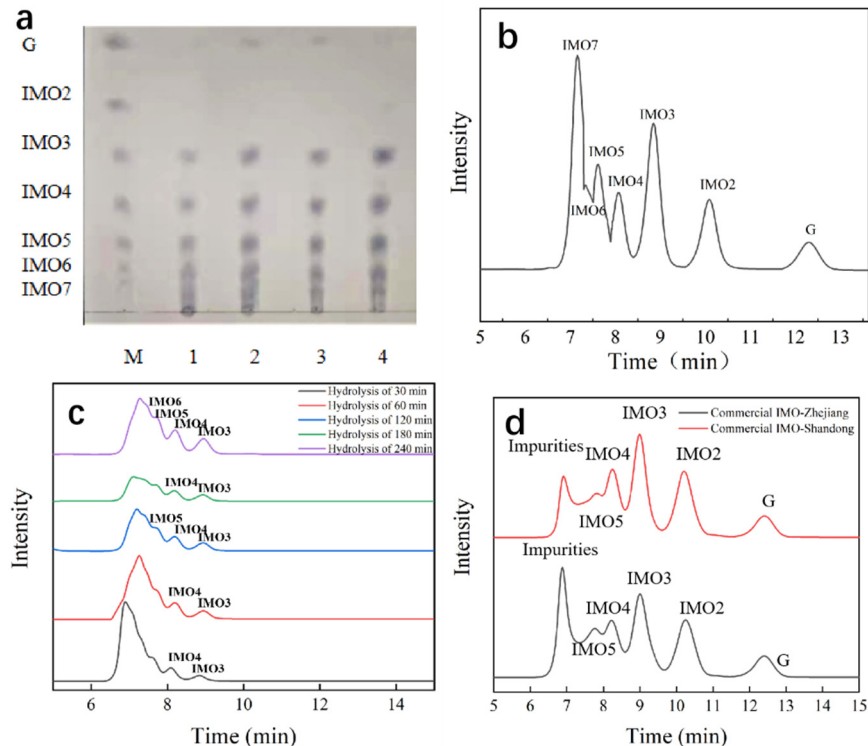

**Figure 4.** Analysis of hydrolysis products of PsDex1711. (**a**) TLC analysis: M (standard: glucose, isomaltose—heptagosaccharide); 1–4: PsDex1711 reacted with dextran T70 for 0.5, 1, 2, and 3 h, respectively; (**b**–**d**) HPLC analysis; (**b**) standard substance: glucose, isomaltodisaccharide—heptagosaccharide; (**c**) PsDex1711 reacted with dextran T70 for 0.5, 1, 2, 3, and 4 h, respectively; (**d**) Composition analysis of two types of isomaltose: CIMO-Zhejiang and CIMO-Shandong.

**Table 2.** Composition analysis of PsDex1711 hydrolysates.

| Duration of Hydrolysis | Proportion of Hydrolysates (%) | | | |
|---|---|---|---|---|
| | IMO3 | IMO4 | IMO5 | IMO6 |
| 30 min | 33.45% | 66.55% | 0 | 0 |
| 60 min | 35.72% | 64.28% | 0 | 0 |
| 120 min | 22.34% | 37.16% | 40.50% | 0 |
| 180 min | 26.32% | 24.50% | 49.18% | 0 |
| 240 min | 12.50% | 11.63% | 19.05% | 56.90% |

This data has been removed from the substrate residues and the percentage is calculated as the peak area of IMO.

### 2.5. Analysis of Antioxidant Activity of the PsDex1711 Hydrolysate

As shown in Figure 5, when the concentration of the PsDex1711 hydrolysate was 40–100 mg/mL, the DPPH clearance rate was approximately 60–70%. The PsDex1711 hydrolysate exhibited high scavenging activity for the hydroxyl radical. When the concentration was 100 mg/mL, it exhibited 80% scavenging effect, which is approximately equal to the scavenging effect of 400 μg/mL VC. The scavenging rate of superoxide anion was approximately 60–70%, which is equivalent to the scavenging effect of 7.5 μg/mL VC. The reducing force was approximately 5 μg/mL VC. Compared with the nonclonal expression, the hydroxyl radical scavenging activity and superoxide anion scavenging rate were greatly improved by 40% and 30%, respectively.

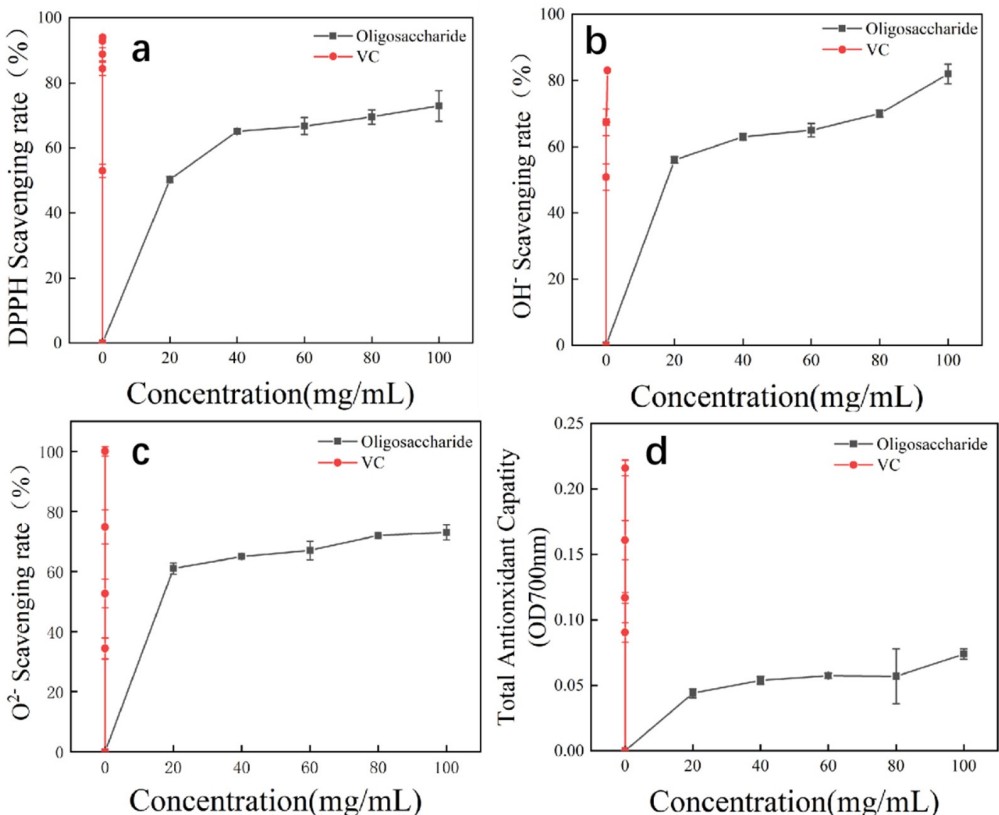

**Figure 5.** Antioxidant activity analysis of hydrolysis products of PsDex1711. The red line is the amount of VC added. (**a**) DPPH scavenging rate; (**b**) Hydroxyl radical scavenging activity; (**c**) Superoxide anion clearance rate; (**d**) Antioxidant capacity.

### 2.6. Effects of IMOs on the Growth of Probiotics

The lyophilized hydrolysis products of the 240 min interaction between PsDex1711 and dextran T70 were added to MRS medium at different concentrations (0.5, 1, and 2.5 g/L) and inoculated with four types of probiotic bacteria to observe their growth, as shown in Figure 6. The promoting effect of IMOs on the growth of the four types of probiotics was related to IMO concentration. The concentration of 1 g/L exhibited the best promotion effect on *Lactobacillus* and *B. lactis*, and the concentration of 2.5 g/L has the best promotion effect on *B. infantis*; too low the promoting effect is not obvious, too high the concentration inhibits the growth of probiotics. At the same time, the promoting effect was related to the culture time. No significant difference in the promoting effect of IMOs was noted before 12 h, but a significant difference appeared at 12 h, indicating that IMOs had an effect on the strain's growth at the later stage of culture. The promotion effect of the two commercial isomaltooligosaccharides was not as obvious as that of the PsDex1711 hydrolysate. Moreover, the addition level was too high to inhibit the growth of probiotics. However, the promoting effect of CIMO-Z was lower than that of CIMO-S possibly because CIMO-Z has more impurities, which also proves the promoting effect of IMOs.

Furthermore, the composition of PsDex1711 hydrolysates was analyzed. PsDex1711 hydrolysates were mainly IMOs with a high DP (IMO4, IMO5, and IMO6), whereas the commercial isomaltooligosaccharides were mainly IMOs with a low DP (IMO2 and IMO3). IMOs with a high DP are possibly more conducive to the absorption of probiotics and thus to their growth.

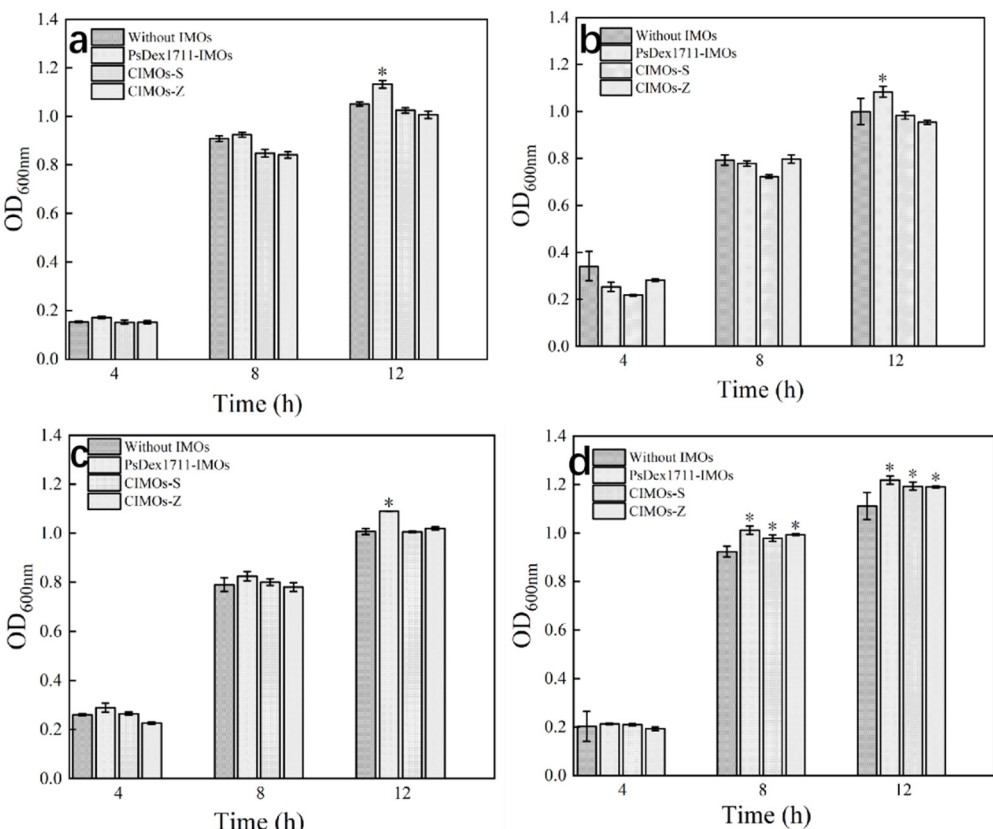

**Figure 6.** Effects of different isomaltose supplemental levels on the growth of the four probiotics: IMO supplemental levels were 0.5, 1, and 2.5 g/L, respectively; (**a**) *Lactobacillus paracei*, IMO was 1 g/L; (**b**) *Bifidobacterium infantis*, IMO was 2.5 g/L; (**c**) *Lactobacillus reuteri*, IMO was 1 g/L; (**d**) *Bifidobacterium lactis*, IMO was 1 g/L. *: The results were significantly different compared to the control group.

## 2.7. Promoting Effect of IMOs on Probiotic Biofilm Formation

Probiotics produce adhesives such as lipoteichoic acid, S-layer protein, lipopolysaccharide, and peptidoglycan to increase their adhesion to intestinal epithelial cells. This adhesion helps them colonize the intestine, inhibit the colonization of pathogenic bacteria, and improve body immunity [22]. *B. lactis* exhibiting the best growth promotion effect of IMOs was selected to explore the promotion effect on biofilm formation. The results are shown in Table 3: the addition of different types of IMOs can accelerate the formation of lactic acid bacteria biofilm, while the addition of the PsDex1711 hydrolysate had the best promoting effect on biofilm formation, reaching 58.82%, which is approximately 8% higher than the effect observed with the two commercial isomaltooligosaccharides.

**Table 3.** Promoting effect of IMOs on formation of probiotic biofilm.

| Add the Type of IMOs | Biofilm Promotion Rate (%) |
|---|---|
| Control (0) | 0 |
| Add 1 g/L PsDex hydrolysate | 58.82 ± 0.20 |
| Add 1 g/L CIMO-S | 50.05 ± 0.26 |
| Add 1 g/L CIMO-Z | 50.20 ± 0.68 |

Scanning electron microscopy observation is presented in Figure 7. When no IMOs were added, *B. lactis* was observed to form a thin monolayer biofilm with low bacterial load on the slide surface, while with the addition of IMO, the bacterial load increased and the biofilm became dense and thickened to a multilayer biofilm. The possible reason is that with the addition of a certain amount of IMOs, the secretion of S-layer proteins and various

extracellular polysaccharides is promoted, thus increasing the adhesion and aggregation ability of lactic acid bacteria, promoting the formation of its biofilm, and contributing to its colonization in the intestinal tract.

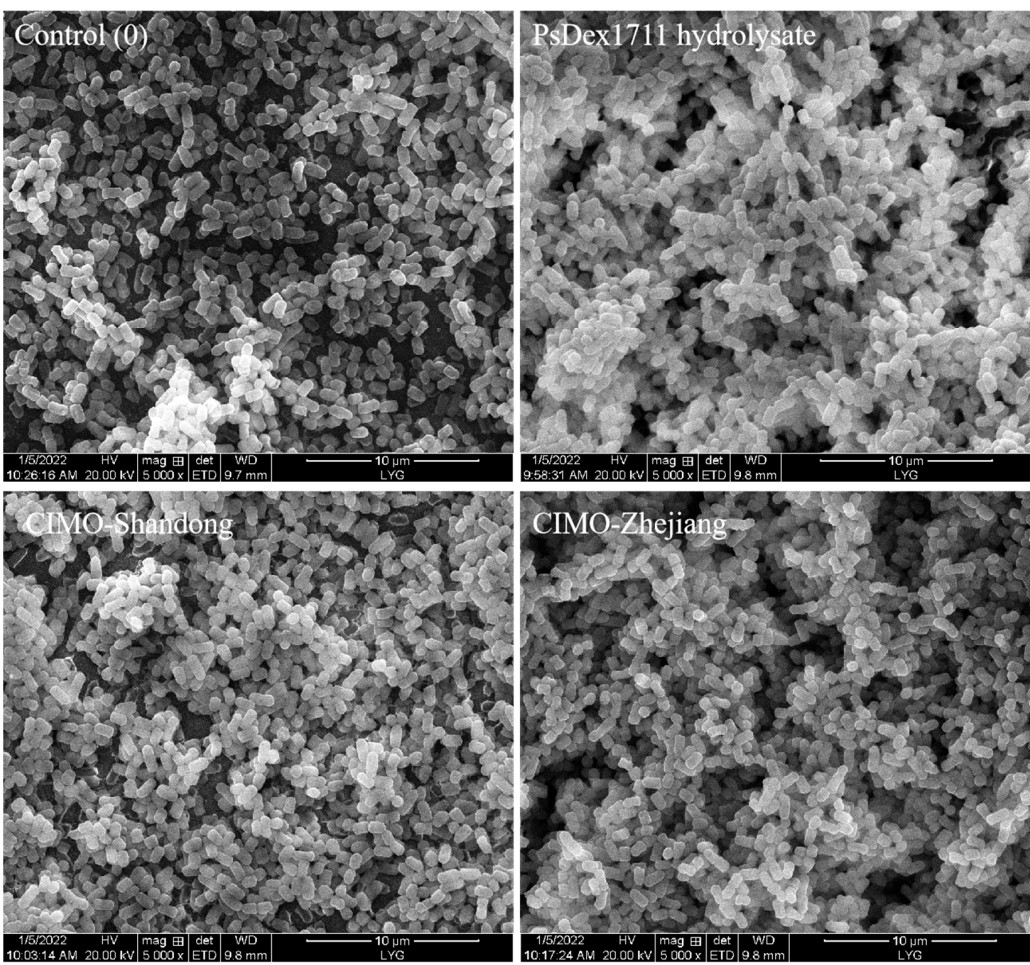

**Figure 7.** The promoting effect of IMOs on the formation of lactic acid bacteria biofilm was observed through scanning electron microscopy. (5000×, from the left to right with no addition, add the PsDex1711 hydrolysate, add CIMO-S, and add CIMO-Z).

## 3. Discussion

AoDex (6nzs.1.A) belongs to the GH49 family of dextranase (EC 3.2.1.11), and the structure of PsDex1711 is similarity with it based on homology modeling. A cleft is on the surface of the C-terminal structural domain in which the predicted catalytic domain (Gln402—Asp424) is located [10]. In contrast, the structure of GH66 family dextranases, which are also endo-dextranases, is different. the SmDex of GH66 family consists of three main structural domains, namely the N-terminal (22—99aa), C-terminal (733—850aa) and A-terminal catalytic structural domain (100—732aa) [23]. The different structural features have a great influence on their catalytic properties and product types, and the homology modeling provides a basis for the subsequent molecular modification of PsDex1711 and specific structural studies.

*E. coli* was the first host bacterium used for recombinant protein production. It has a clear genetic background, simple operation, and high transformation and transduction efficiency, grows fast, is a low-cost host, can quickly produce the target protein at a large scale, and can express considerably higher levels of the exogenous gene product than other genes, amount of target protein expression can even more than 30% of the total amount of protein [24]. *E. coli* is therefore the most widely used protein expression system.

The molecular weight of Dextranase protein is closely related to its source. Pavel V. Volkov et al. [9] cloned and expressed dextran with a molecular weight of 60–63 kDa from *Penicillium funiculosum*. J.J. Virgen-Ortiz et al. [25]. found that the molecular weight of the purified dextranase gene from *Chaetomium erraticum* was 59 kDa, which was approximately 120 kDa through enzyme spectrum analysis. Elvira Khalikova et al. [14] screened for a dextran anhydrase strain and purified it. Three isomers having molecular weights of 76, 89, and 110 kDa were formed.

The optimal catalytic temperature and pH of PsDex1711 were 30 °C and 6.0, respectively. After cloning and expression, the optimal temperature decreased to 30 °C, and the enzyme activity was high in the temperature range of 15–40 °C. The enzyme activity decreased sharply with an increase in temperature, exhibiting good cold adaptability. As an enzyme that can efficiently catalyze at a low temperature and quickly inactivate at a high temperature, cryogenic enzymes have unique advantages in biotechnological applications [26]. It also has a potential wide application in the cold washing and food additives industry, bioremediation of contaminated environments, biotransformation, and molecular biology research. However, the thermal stability of enzymes was one of the key properties to affect their industrial applications. Researchers could more accurately locate the key amino acid residues or peptides that affected the thermal stability of enzyme molecules through the structural parameters (e.g., B-factor, RMSF value, etc.) and then targeted mutations were used to improve the thermal stability of enzymes [27–29].

The optimal pH of PsDex1711 was 6.0, and the enzyme activity was >80% in the pH range of 5.0–9.0. The enzyme activity was higher in the pH range of 5–6. *Streptococcus mutans* is the most important bacterium causing caries and can easily grow under acidic conditions [30]. Thus, PsDex1711 has a crucial research value in the prevention and treatment of caries.

PsDex1711 hydrolyzes dextran T70 to produce IMO3, IMO4, IMO5, and IMO6, but no glucose was produced. PsDex1711 is an endodextranase that hydrolyzes the $\alpha$-(1,6) bond in dextran from the inside to reduce its molecular weight. In contrast, exodextranase cut outward from the reducing end to release glucose [31]. Most dextran anhydrases reported are endotangent dextran anhydrases, and the product formed (i.e., IMOs) has a crucial application value.

In this study, the antioxidant activity of the hydrolysate was first analyzed. The PsDex1711 hydrolysate exhibited a good scavenging effect on DPPH, hydroxyl radicals and superoxide anions. Their $IC_{50}$ values were 20, 20, and 15 mg/mL, respectively. Dong Dongxue et al. [17] analyzed the antioxidant activity of hydrolysates that produced by GH66 family dextranase BaDex, and the scavenging of DPPH and superoxide anion were 40% and 66.8% at 40 mg/mL. Junpeng Yi et al. [32] extracted polysaccharides from *Achyachyra vulgaris* and conduced steam explosion treatment. The scavenging rates of DPPH, hydroxyl radicals, and superoxide anions at $IC_{50}$ were 0.125, 0.5, and 0.25 mg/mL, respectively.

The PsDex1711 hydrolysate had a significant promoting effect on the growth of probiotics. The main components of IMOs are isomaltose, panlose, isomaltotriose, and more than tetrasaccharide. These components are bound by $\alpha$-(1,6) glucosidic linkages between glucose molecules. Isomaltose can promote the proliferation of intestinal bifidobacteria, inhibit the formation of harmful intestinal bacteria and substances, increase vitamin content, and improve body immunity [33]. Isomaltose is not absorbed by the stomach and small intestine, but directly enters the large intestine, which is preferentially used by bifidobacteria to help multiply in large numbers. Isomaltose is a bifidobacteria proliferation factor. However, other harmful intestinal bacteria cannot utilize isomaltose, thus inhibiting the growth of harmful bacteria and promoting the adjustment of intestinal microecology to a virtuous cycle [34]. On adding 0–8 mg/mL of algae polysaccharide into the plate of *L. rhamnosus* culture, Zhu Yanli et al. [35] found that the colony count of these bacteria increased from $3 \times 10^9$ CFU/mL to $5 \times 10^9$ CFU/mL. By adding fructose-oligosaccharide to yogurt, Guilherme M et al. [36] found that the number of lactic acid bacteria increased

by 104 CFU/mL compared with that in the control group after storage for some time; this result is similar to that of the present study.

At the same time, IMOs can also promote biofilm formation by lactic acid bacteria and contribute to the colonization of these bacteria in the intestinal tract. This colonization is mainly related to the adhesion ability of the strain, its motility, lactic acid secretion, and its proliferation ability [37]. According to this study, IMOs can be added to promote the growth of lactic acid bacteria and may simultaneously induce the expression of S-protein and the secretion of extracellular polysaccharides such as lipopolysaccharides, thus promoting biofilm formation. Michał Piotrowski et al. [38] found that the addition of salivated breast milk oligosaccharide (HMO) inhibits the expression of gene encoding the cell wall protein (Cwp) 84 in *Clostridium difficile*, thus inhibiting biofilm formation by these bacteria.

In the present study, the dextranase PsDex1711 mainly produced isomaltose (degree of polymerization, DP, DP4, or above), which could improve probiotic activity and thus has more research value. Fatheema B. Subhan et al. [39] found that IMOs possess new characteristics, and when administered as an oral solution, IMOs with a high degree of aggregation stimulate the secretion of insulin and incretin without increasing blood sugar levels. Jarunee Kaulpiboon et al. [40] used cassava starch to prepare a high DP IMO. Compared with a low DP IMO, a high DP IMO can better promote probiotic growth. Moreover, the $OD600_{nm}$ of a high DP IMO was 2.2 after 24 h culture, whereas it was only approximately 1.0 for a low DP IMO. Hu Ying et al. [41] found that a high DP IMO was less easily digested by gastric juices than a low DP IMO, thereby allowing it to reach the intestine to stimulate the growth of probiotics. Therefore, a high DP IMO has more research value in food, medical, and other industries. In the future, we intend to study the structure of PsDex1711, identify the mechanism that affects the type of hydrolysate, determine the site, and carry out site-directed mutations to obtain a high DP IMO.

## 4. Materials and Methods

### 4.1. Material

*Pseudoarthrobacter* sp. RN22 was screened from the sea mud near Rizhao by our laboratory. The strain has been conserved in the China General Microbiological Culture Collection Center, with the conservation number CGMCC No. 19740. The bacterial genome extraction kit, plasmid microextraction kit, generic agarose gel DNA recovery kit, and competent *Escherichia coli* DH5$\alpha$ and *E. coli* BL21 (DE3) cells were purchased from Tiangen Biochemical Technology Co., Ltd. De Man Rogosa Sharpe medium (MRS medium) were purchased from Beijing Luqiao Technology Co., Ltd. (Beijing, China). *Lactobacillus paracasei* JLPF-176, *L. reuteri* JYLB-131, *Bifidobacterium lactis* JYBR-390, and *B. infantis* JBLC-141 were purchased from Shandong Zhongke Jiayi Biological Engineering Co., Ltd. (Weifang, Shandong, China). Two types of commodity IMOs were purchased from Shandong Bailong Chuangyuan Biotechnology Co., Ltd. (Dezhou, Shandong, China) and Zhejiang Yinuo Biotechnology Co., Ltd. (Hangzhou, Zhejiang, China) Restriction enzymes NdeI and SacI were purchased from New England Biolabs (Beijing) Ltd. (Beijing, China). The pET29a plasmid was preserved in our laboratory. Other reagents were of analytical grade.

### 4.2. Culture Conditions

The strain *Pseudoarthrobacter* sp. RN22 was cultured in 2216E medium (5 g/L fish meal peptone and 1 g/L yeast powder, prepared with aged seawater, pH 8.0). The recombinant plasmid was cultured in Luria Broth medium (LB medium) (10 g/L tryptone, 5 g/L yeast powder, 10 g/L NaCl, and kanamycin; final concentration of 50 μg/mL). *L. paracasei*, *L. reuteri*, *B. lactis*, and *B. infantis* were cultured in MRS medium (10 g/L peptone, 10 g/L beef powder, 5 g/L yeast powder, 20 g/L glucose, 0.5 g/L $MgSO_4$, 5 g/L $CH_3COONa$, 2 g/L$(NH_4)_2HC_6H_5O_7$, 1 mL/L Tween 80, 0.25 g/L $MnSO_4$, 2 g/L $K_2HPO_4$, pH 6.2–6.4).

### 4.3. Bioinformatics Analysis and Homology Modeling of the PsDex1711 Gene Sequence

The PsDex1711 gene was sequenced from the whole genome of the strain *Pseudoarthrobacter* sp. RN22. The gene sequence was submitted to the NCBI database, the sequence accession number is PRJNA837017. Blast online comparison was used, and similar sequences were downloaded. The phylogenetic tree was constructed with Mega 7 software by using the neighbor joining method, and the signal peptide of the amino acid sequence was analyzed using SignalP-5.0 online software. The physical and chemical properties were analyzed using ExPASy online software, the transmembrane domain was analyzed using TMHMN server online software, and the hydrophilicity and hydrophobicity of the protein were analyzed using DNAMAN V6 software. The protein secondary structure was predicted using SOPMA online software. The CDD database in the NCBI database was used to analyze the protein's conservative domain. SWISS-MODEL was used for homology modeling, and AoDex (PDB Code: 6nzs.1. A) was used as the template. Procheck, Errat, and Verify 3D were used to detect model rationality. The results were presented using PyMOL software.

### 4.4. PCR Amplification and Cloning Vector Construction

The glycerol-preserved *Pseudoarthrobacter* sp. RN22 strain was inoculated in 2216E medium and cultured at 37 °C and 180 r/min for 12 h. The DNA was extracted using the bacterial genome extraction kit. Primers were designed according to the PsDex1711 gene sequence and pET29a plasmid gene sequence: PsDex1711-F: 5′-CGCCATATGATGAAGCATT-ACCTCC-3′, PsDex1711-R: 5′-CGAGCTCCCACGCGTTCCAGGTAT-3′. The target gene was amplified using Taq PCR Master Mix. The *E. coli*-pET29a plasmid preserved in glycerol was inoculated into 50 μg/mL kanamycin LB liquid medium and cultured overnight at 37 °C and 180 r/min. The plasmid was extracted using the plasmid small extraction kit.

The restriction endonucleases NdeI and SacI were used to simultaneously digest the target gene PsDex1711 and pET29a plasmid. The reaction was performed for 8–10 h at 37 °C, and 1% agarose gel was used to detect and recover the target gene and pET29a. The target gene was linked to the vector pET29a with solution I. After reaction at 16 °C for 1 h, the vector was transferred to *E. coli* DH5α competent cells. Single colonies on the plate were sent to Sangong Bioengineering (Shanghai) Co., Ltd. (Shanghai, China) for sequencing, and the sequencing results were compared with the sequence of the target gene PsDex1711. After the correct clones were cultured, seeds were preserved, and plasmids were extracted.

### 4.5. Expression and Purification of Recombinant Plasmid

The extracted pET29a-PsDex1711 plasmid was transferred into BL21 (DE3) competent cells. Single colonies were inoculated into LB liquid medium containing 50 μg/mL kanamycin and incubated at 37 °C and 180 r/min. When OD600 nm was 0.6–0.8, 4% of the culture was inoculated in the new LB liquid medium containing kanamycin 50 μg/mL and incubated at 37 °C and 180 r/min. Then, 0.1, 0.5, and 1 mM isopropyl-β-D-thiogalactoside (IPTG) was cultured at 16 °C, 24 °C, and 48 °C for 24 and 48 h at 180 r/min, respectively.

The cultured bacterial solution was centrifuged at 8000 r/min for 15 min and the supernatant was discarded. The precipitated bacteria were cleaned twice with 0.01 M PBS buffer (g/L: 0.27 $KH_2PO_4$, 1.42 $Na_2HPO_4$, 8 NaCl, 0.2 KCl, pH 7.4), and the bacterial cells were resuspended with PBS buffer of one-tenth of the initial bacterial solution volume. The cells were broken by ultrasound for 15 min and centrifuged at 8000 r/min for 10 min. The supernatant was collected to obtain crude enzyme solution. With dextran T20 as the substrate, enzyme activity was determined using the DNS method at 45 °C. The relative enzyme activity was calculated to obtain the best expression of PsDex1711.

The crude enzyme solution containing His-tag was purified through nickel column affinity chromatography. The crude enzyme solution (5 mL) was combined in the nickel column for 1 h to obtain the effluent, and then imidazole-tris HCl buffer at 20, 40, 60, 80, 100, 200, and 300 mM was used successively (final concentration of Tris HCl and NaCl was 50 and 300 mM, respectively, and 0.22 μm membrane filtration sterilization was performed)

to elute the protein in the nickel column. The protein concentration of each eluent was determined using the BCA protein concentration determination kit and detected through 10% SDS-PAGE vertical electrophoresis.

### 4.6. Determination of Enzyme Activity by the DNS Method

At a certain temperature and pH, 150 μL of 3% dextran T70 was added to 50 μL enzyme solution. After allowing the reaction to continue for 15 min, 200 μL DNS was added to terminate the reaction. The mixture was placed in boiling water bath for 5 min and then placed in ice water to terminate the reaction. Then, 3 mL ultrapure water was added to dilute the mixture. The $OD_{540\,nm}$ value of 200 μL of the mixture is measured to determine the amount of reducing sugar. DNS was added first, followed by the addition of the enzyme solution as the control group [42]. The enzyme activity was calculated according to the following formula:

$$\text{Dextranse activity (U/mL)} = \frac{\text{Reducing sugar quality (μg)} \times \text{Dilutionmulitiple}}{\text{Glucose molecular weight (g/mol)} \times \text{responsetime (min)} \times \text{Enzymeliquid product (mL)}} \quad (1)$$

unit definition of enzyme activity (U/mL): amount of enzyme solution required for hydrolysis of dextran to release 1 μmol of isomaltose per minute.

### 4.7. Enzymatic Properties of PsDex1711

#### 4.7.1. PsDex1711 Optimum Catalytic Substrate

Using 3% dextran of different molecular weights (T20, T40, T70, and T500), chitosan, pullulan polysaccharide, and soluble starch as substrates, the enzyme activity was measured at 45 °C. All experiments were performed in triplicate.

#### 4.7.2. Optimum Catalytic Temperature and Temperature Stability of PsDex1711

The enzyme activity was measured at different temperatures (30 °C, 40 °C, 45 °C, 50 °C, 55 °C, and 60 °C) and the optimal substrate. The residual enzyme activity was measured after holding the enzyme at 30 °C, 40 °C, and 50 °C for 20, 40, 60, 100, and 120 min. All experiments were performed in triplicate.

#### 4.7.3. PsDex1711 Optimum Catalytic pH and pH Stability

The buffer solution (pH 4.0–9.0, final concentration of 50 mM) was mixed with the substrate, and the enzyme activity was measured at the optimum temperature. The enzyme solution was mixed with buffer solutions at different pH and a final concentration of 50 mM, and the residual enzyme activity was determined at 30 °C for 1 h. All experiments were performed in triplicate.

#### 4.7.4. Determination of Enzyme Kinetic Constants of PsDex1711

The substrate dextran T500 was prepared at different concentrations as 1.2, 3, 4, 4.5, 5, 6, 7, 8, 10, 20, and 30 mg/mL. One hundred fifty microliter of substrate was mixed with 50 μL of purified PsDex1711 enzyme solution, and the enzyme reaction rate was measured under the optimum temperature and pH conditions. The Km and Vmax values were calculated by the double inverse plotting method (Lineweaver-Burk) with 1/[S] as the horizontal coordinate and 1/V as the vertical coordinate, with the slope of Km/Vmax and the intercept of 1/Vmax.

### 4.8. Component Analysis of PsDex1711 Hydrolysate

The PsDex1711 enzyme solution was mixed with the substrate dextran T70 1:3 and incubated at 30 °C for 0.5, 1, 2, 3, and 4 h, placed in a boiling water bath for 5 min, and centrifuged at 8000 r/min for 5 min. The supernatant was filtered using a 0.22-μM membrane and stored at 4 °C.

Isomaltose (IMO2), isomaltotriose (IMO3), isomaltotetraose (IMO4), isomaltopentaose (IMO5), isomaltohexaose (IMO6), and isomaltoheptaose (IMO7) were used as the standard

for thin layer chromatography (TLC). All samples and commercial isomaltooligosaccharide were taken at 3 μL for TLC analysis.

For HPLC analysis, the chromatographic column from Waters Sugar-Pak-1 was used. The mobile phase was deionized water, the flow rate was 0.4 mL/min, and the injection volume was 20 μL. The detector is a differential refractive index detector. The composition and content of each standard and sample were analyzed on the basis of reported peak areas.

### 4.9. Component Analysis of PsDex1711 Hydrolysate

#### 4.9.1. DPPH Radical Scavenging Rate

PsDex1711 and dextran T70 were mixed at a ratio of 1:3, reacted at 30 °C for 4 h to obtain the hydrolysate, lyophilized for 24 h, and configured into hydrolysates of different concentrations (20, 40, 60, 80, and 100 mg/mL). Then, 200 μL of PsDex1711 hydrolysates of different concentrations was mixed with 0.1 mM DPPH (absolute ethanol configuration). The mixture was kept away from light for 30 min, centrifuged at 5000 r/min for 10 min, and measured to determine the $OD_{517nm}$ value [43]. Ascorbic acid (VC) at different concentrations (2.5, 5, 10, and 50 μg/mL) was used as a reference. The formula for determining the DPPH radical scavenging rate is as follows:

$$w(\%) = \left(1 - \frac{Ai - Aj}{A0}\right) \times 100\% \tag{2}$$

*Ai*: absorbance of the reaction between DPPH and the PsDex1711 hydrolysate; *Aj*: absorbance of the reaction between absolute ethanol solution and the PsDex1711 hydrolysate; *A*0: absorbance of DPPH and water

#### 4.9.2. Hydroxyl Radical Scavenging Activity

First, we mixed 100 μL of 9 mM $FeSO_4 7H_2O$, 100 μL of 9 mM salicylic acid (prepared with absolute ethanol), 100 μL of different concentrations of the PsDex1711 hydrolysate, and 100 μL of 0.03% $H_2O_2$; incubated this mixture at 37 °C in dark for 15 min and centrifuged at 5000 r/min for 10 min. Then, the $OD_{510nm}$ value was measured [44]. Different concentrations (20, 50, 100, and 400 μg/mL) of VC were used as the reference. The formula for calculating hydroxyl radical scavenging activity is as follows:

$$w(\%) = \frac{Aj - An}{Aj - An} \times 100\% \tag{3}$$

*Ai*: PsDex1711 hydrolysate; *Aj*: control without the PsDex1711 hydrolysate; *An*: a blank reagent.

#### 4.9.3. Superoxide Anion Scavenging Activity

Here, 200 μL of the PsDex1711 hydrolysate was mixed with 1 mL Tris HCl (50 mM) and incubated at 25 °C for 10 min. Then, pyrogallol (6 mM) was added to this mixture. The mixture was placed under room temperature for 30 min, and its $OD_{320nm}$ value was measured [45]. Different concentrations (2.5, 5, 7.5, and 10 μg/mL) of VC were used as the reference. The formula for calculating superoxide anion scavenging activity is as follows:

$$w(\%) = \left(1 - \frac{Ai - Aj}{A0}\right) \times 100\% \tag{4}$$

*Ai*: the hydrolysate of PsDex1711 + Tris-HCl + pyrogallol; *Aj*: the hydrolysate of PsDex1711 + Tris-HCl + HCl; *A*0: water + Tris-HCl + pyrogallol

#### 4.9.4. Determination of Reducing Power

A mixture of 100 μL PsDex1711 hydrolysate, 100 μL of 0.2 M PBS (pH 6.6), and 100 μL of 1% $K_3Fe (CN)_6$ was prepared and incubated at 50 °C for 20 min. Then, 100 μL of 10% TCA was added and the mixture was centrifuged at 5000 r/min for 10 min. To the

supernatant, add 1.25 mL of distilled water with 50 μL of 0.2% $FeCl_3$ and measure the $OD_{700nm}$ value [46]. Different concentrations (5, 10, 20, and 40 μg/mL) of VC were used as the reference.

### 4.10. Effects of IMOs on the Growth of Probiotics

The PsDex1711 enzyme solution was mixed with the substrate T70 1:3 and incubated at 30 °C for 240 min. The mixture was freeze-dried for 24 h to obtain IMOs. IMOs and the commercial isomaltooligosaccharide were added to MRS medium at a concentration of 0.5, 1, and 2.5 g/L.

*L. paracasei*, *L. reuteri*, *B. lactis*, and *B. infantis* were inoculated in MRS medium at 1% inoculation amount, and the seed solution was prepared through anaerobic culture at 37 °C for 12 h. The activated seed solution was inoculated in MRS medium with different content of IMOs at 1% inoculation amount and incubated under static anaerobic conditions at 37 °C. The $OD_{600nm}$ value of the activated seed solution was measured at 200 μL every 4 h, and MRS medium without IMOs was used as a control. The data were analyzed using IBM SPSS statistical software. The significant difference was determined using the independent sample *t*-test, and the mean value at 95% significance level was compared ($p < 0.05$).

### 4.11. Effects of IMOs on the Formation of Probiotic Biofilms

The strain with the best promoting effect of IMO in experiment 2.10 and the corresponding IMO supplemental amount were selected for the biofilm formation experiment. The strain was inoculated (1%) in MRS medium, and the seed liquid was prepared through anaerobic culture at 37 °C for 12 h.

First, 190 μL MRS medium containing different IMO types was added to the 96-well plate. Then, 10 μL seed solution was inoculated in the MRS medium present in the plate and incubated at 37 °C for 13 h. The redundant bacterial solution was removed, the well wall was rinsed with PBS buffer twice (5 min each), and the well plate was dried for 15 min. Then, 100 μL of 2.5% glutaraldehyde was added and fixed for 15 min. Finally, 220 μL of 0.1% crystal violet (a dye) was added to each well for 5 min. The dye solution was discarded. The biofilm was washed with slow water immediately and dried. Then, the amount of biofilm formation was measured using an enzyme plate tester and expressed as $OD_{595nm}$ value [47]. All experiments were performed in quadruplicate; MRS medium without IMOs was used as the control.

### 4.12. Scanning Electron Microscopy for Observing the Effect of IMOs on Probiotic Biofilm Formation

The slide was placed in a 24-well plate. Then, 1 mL of 2% gelatin was added and the plate was left overnight. MRS medium (1.9 mL) containing different IMO types was added. Then, 100 μL of seed solution was inoculated and cultured at 37 °C for 13 h. The bacterial solution was removed and washed with water twice, fixed with 2.5% glutaraldehyde at 4 °C for 4 h, and dehydrated with 50%, 70%, 80%, 90%, and 100% alcohol. After drying, the sample was sent to the National Silicon Inspection Center for observation. All experiments were performed in duplicate; MRS medium without IMOs was used as the control.

## 5. Conclusions

In summary, we cloned and expressed the PsDex1711 gene of *Pseudarthrobacter* sp. RN22, which is 1860-bp long and encodes 620 amino acids. The conserved domain analysis of Psdex1711 showed that the N-terminal domain contained 192 amino acids (Gly 33-THR 225) and the C-terminal domain contained 125 amino acids (Gly 495-TRP 620). The optimal expression conditions were as follows: final IPTG concentration of 0.5 mM and induction at 16 °C for 24 h. The enzyme was purified using a nickel column and its molecular weight was approximately 70 kDa. The optimum pH of the enzyme was 6.0 and the optimum temperature was 30 °C. The enzyme hydrolyzed dextran T70 to produce IMO3, IMO4, IMO5, and IMO6. The PsDex1711 hydrolysate had a certain antioxidant effect. At 1 and

2.5 g/L, the PsDex1711 hydrolysate significantly promoted the growth of *Lactobacillus* and *Bifidobacterium* after 12 h culture, and the biofilm formation of *Lactobacillus* and *Bifidobacterium* was promoted by 58.82%. Thus, the recombinant dextranase PsDex1711 and its product IMOs have broad application prospects in food, medical, and other industries.

**Supplementary Materials:** The following supporting information can be downloaded at: https://www.mdpi.com/article/10.3390/catal12070784/s1, Figure S1: Evolutionary tree analysis of PsDex1711; Figure S2: Expression conditions of PsDex1711. (a) The effect of IPTG concentration on PsDex1711 expression; (b) The effect of temperature on PsDex1711 expression; (c) The effect of time on PsDex1711 expression; Table S1: Analysis of different components in commercial isomaltose.

**Author Contributions:** H.W., Conceptualization, methodology, and writing—original draft preparation. Q.L. and D.D.: data curation and resources. Y.X. and M.L. (Mingwang Liu); visualization and investigation. J.L.; software and validation. H.W.; writing—reviewing and editing. M.L. (Mingsheng Lyu) and S.W.; funding acquisition, project administration. All authors have read and agreed to the published version of the manuscript.

**Funding:** This study was supported by the National Natural Science Foundation of China (Grant no. 32172154); 521 Program Grant No. LYG06521202107; The Priority Academic Program Development of Jiangsu Higher Education Institutions (PAPD); The Research and Practice Innovation Program of Jiangsu (KYCX2021-020).

**Institutional Review Board Statement:** This article does not contain any studies with human participants or animals performed by any of the authors.

**Data Availability Statement:** The datasets generated during and/or analyzed during the current study are available from the corresponding author on reasonable request.

**Conflicts of Interest:** The authors declare no conflict of interest.

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
