# Peer review of "Cloning of Cold-Adapted Dextranase and Preparation of High Degree Polymerization Isomaltooligosaccharide"

_catalysts, doi:10.3390/catal12070784_

Round 1
Reviewer 1 Report
The manuscript reports the production of a recombinant cold-adapted dextranase from a marine bacterium - Pseudarthrobacter sp. RN22- and produced in E. coli as a host. The enzyme was characterized in terms of optimal pH, stability at different temperatures, and product profile by using as a substrate dextran. The products were isomalooligosaccharides, which were tested as potential prebiotics for Lactobacillus and Bifidobacterium strains. Their antioxidant properties were also assessed. In summary, this work aims to propose the dextranase PsDex1711 as a catalyzer to produce isomaltooligosaccharides with prebiotic and antioxidant properties. Before publication some clarifications are needed.
- In the abstract is stated that the dextranase was expressed in Pseudarthrobacter sp. RN22. However, in material and methods is described the production of the recombinant enzyme in E. coli BL21(DE3).
- The kinetic characterization is missing. As it is a new enzyme, a complete characterization is needed.
- Although a molecular model was claimed in the results, there is no discussion on the model. It is not clear the purpose of obtaining the model.
Author Response
Dear Reviewers,
We are really appreciated for your kindly consideration to give us an opportunity to revise our manuscript entitled “Cloning of cold-adapted dextranase and preparation of high degree polymerization isomaltooligosaccharide” (Catalysts-1805211). We appreciate reviewers for the comments, and we have amended our manuscript according to the comments carefully. We have tried our best to revise our manuscript. Revised portions have been marked with yellow background in the manuscript. The main corrections in the paper and the responds to the reviewer’s comments are as following:
Responds to the reviewer 1’s comments:
The manuscript reports the production of a recombinant cold-adapted dextranase from a marine bacterium - Pseudarthrobacter sp. RN22- and produced in E. coli as a host. The enzyme was characterized in terms of optimal pH, stability at different temperatures, and product profile by using as a substrate dextran. The products were isomalooligosaccharides, which were tested as potential prebiotics for Lactobacillus and Bifidobacterium strains. Their antioxidant properties were also assessed. In summary, this work aims to propose the dextranase PsDex1711 as a catalyzer to produce isomaltooligosaccharides with prebiotic and antioxidant properties. Before publication some clarifications are needed.
- In the abstract is stated that the dextranase was expressed in Pseudarthrobacter RN22. However, in material and methods is described the production of the recombinant enzyme in E. coli BL21(DE3).
Response: Thank you very much for the comment. We have revised in the manuscript: In this study, the marine bacterial Pseudarthrobacter sp. RN22 dextranase (PsDex1711) gene was cloned and expressed in E. coli BL21 (DE3). They are in Lines: 14-15.
- The kinetic characterization is missing. As it is a new enzyme, a complete characterization is needed.
Response: Thanks for your comment. We have added the results in the manuscript including to experimental methods, enzyme kinetic constants, and analysis of results. The experimental methods are in lines: 459-466, and the analysis of the results is in lines : 153-157. Thanks.
- Although a molecular model was claimed in the results, there is no discussion on the model. It is not clear the purpose of obtaining the model.
Response: Thanks for your comment. We have added a description and comparative discussion of the model in the manuscript, which was constructed to comprehend the structural features of family types of PsDex1711 and to provide a theoretical basis for future experiments. They are in Lines: 96-105 and 258-267.
We would like to express our great appreciation to you for comments on our manuscript again.
Yours sincerely,
Huanyu Wang
Corresponding author:
Name: Jing Lu & Shujun Wang
E-mail: wanghuanyuhk@163.com

Reviewer 2 Report
Lines 14-15, PsDex1711 gene was cloned in Pseudarthrobacter?, I think there is an error, it must be E. coli.
Lines 265-266, there’s an erro in pH scale
What could be the explanation of the higher antioxidant activity of PsDex1711 hydrolysate versus steam explosion hydrolysate from A. vulgaricus polysaccharides?
Authors must also discuss the bioprocess disadvantages of the enzymatic stability at low temperatures, what would be the strategy to increase thermal stability?.
Author Response
Dear Reviewers,
We are really appreciated for your kindly consideration to give us an opportunity to revise our manuscript entitled “Cloning of cold-adapted dextranase and preparation of high degree polymerization isomaltooligosaccharide” (Catalysts-1805211). We appreciate reviewers for the comments, and we have amended our manuscript according to the comments carefully. We have tried our best to revise our manuscript. Revised portions have been marked with yellow background in the manuscript. The main corrections in the paper and the responds to the reviewer’s comments are as following:
Responds to the reviewer 2’s comments:
- Lines 14-15, PsDex1711 gene was cloned in Pseudarthrobacter?, I think there is an error, it must be coli.
Response: Thank you very much for the comment. We have revised in the manuscript: In this study, the dextranase (PsDex1711) gene of marine bacterial Pseudarthrobacter sp. RN22 was cloned and expressed in E. coli BL21 (DE3). They are in Lines: 14-15.
- Lines 265-266, there’s an erro in pH scale.
Response: Thanks for your comment. We have revised in the manuscript: The optimal pH of PsDex1711 was 6.0, and the enzyme activity was >80% in the pH range of 5.0–9.0. It is in Line: 295.
- What could be the explanation of the higher antioxidant activity of PsDex1711 hydrolysate versus steam explosion hydrolysate from vulgaricus polysaccharides?
Response: Thank you very much for the comment. We have revised in the manuscript. The hydrolysates might have high antioxidant activity after purification treatment. W have added a comparison of the antioxidant activity of the untreated hydrolysis product. They are in Lines: 307-310.
- Authors must also discuss the bioprocess disadvantages of the enzymatic stability at low temperatures, what would be the strategy to increase thermal stability?
Response: Thank you very much for the comment. We have revised in the manuscript to discuss the disadvantages of low temperature enzymes and ways to improve their thermal stability. They are in Lines: 289-293.
We would like to express our great appreciation to you for comments on our manuscript again.
Yours sincerely,
Huanyu Wang
Corresponding author:
Name: Jing Lu & Shujun Wang
E-mail: wanghuanyuhk@163.com

Round 2
Reviewer 1 Report
The udated version is available for publication.
This manuscript is a resubmission of an earlier submission. The following is a list of the peer review reports and author responses from that submission.